# Mechanical and Thermal Stress Behavior of a Conservative Proposed Veneer Preparation Design for Restoring Misaligned Anterior Teeth: A 3D Finite Element Analysis

**Shilan Nawzad Dawood** [1,*]🆔, **Abdulsalam Rasheed Al-Zahawi** [1] **and Laith Abed Sabri** [2]🆔

[1] Conservative Department, College of Dentistry, University of Sulaimani, Sulaimani 46001, Iraq; abdulsalam.kudid@univsul.edu.iq

[2] Department of Physics, Case Western Reserve University, Cleveland, OH, 44106, USA; laith.sabri@case.edu

\* Correspondence: shelan.dawood@univsul.edu.iq

**Abstract:** The objective of this study was to evaluate the biomechanical and thermal behavior of a proposed preparation design as a conservative treatment option that aims to preserve both gingival and tooth health structures through a comparative finite element analysis with non-preparation and conventional designs. 3D solid models of laminate veneers with different preparation designs were obtained using cone-beam computed tomography (CBCT) scanning of the maxillary incisor. A 100-Newton load was applied with angulations of 60° and 125° to the longitudinal axis of the tooth to determine the stresses during mastication. In addition, transient thermal analysis was performed to compare the temperature and thermal distribution of the restored tooth models when subjected to thermal loads of 5 °C and 55 °C. Teeth prepared with the proposed design showed lower stress distributions and a repairable failure mode, followed by the non-preparation design, while teeth prepared with the conventional design showed the highest stress concentrations. Furthermore, cold thermal loading yielded larger thermal stress distributions than hot thermal loading, independent of the preparation type, and the effect of temperature changes were within the critical limit near the pulp and dentin regions. Thus, the preparation design geometry affects the long-term success of laminate restoration, and the proposed design yields more uniform and appropriate stress distributions than the other techniques.

**Keywords:** biomechanics; preparation design; finite element analysis; ceramic laminate veneers; cone-beam computed tomography; mechanical and thermal loads

## 1. Introduction

Patient demands for correction and improvement of unaesthetic anterior teeth has become a fundamental aspect of modern dentistry [1]. For many years, the most expected and durable outcomes for correcting esthetic defects of the diastema, chipped, fractured, malformed, and misaligned teeth in visible areas have been achieved with either full-coverage crown restorations or direct composite veneers [1,2]. Crown restorations are invasive and involve excessive removal of sound tooth structures with possible adverse effects on pulp health and periodontal tissues [3]. In contrast, composite veneers suffer from limited longevity and do not offer the best esthetics due to color instability, marginal fractures, and polymerization shrinkage, thereby compromising the esthetic results and necessitating constant preservation and polishing over time [4,5].

Accordingly, with advances in adhesive systems and the development of newest-generation ceramic technology, the alternative approach of using laminate veneers has become a reliable and

commonly-used treatment modality [6]. This approach is currently considered as one of the most desirable, biocompatible, and conservative methods for satisfying a patient's esthetic requirements with minimum damage to tooth structures, by bonding a thin layer of veneer to the tooth surface using adhesive and luting cement [7,8]. The life expectancy of laminate veneers is influenced by multiple factors, such as tooth morphology, porcelain thickness, the type of luting cement and adhesive system, periodontal response, marginal adaptation, functional and parafunctional activities, and the geometry of the preparation [9].

Preparation design is one of the critical factors affecting the ultimate success and maintenance of veneer restorations. The choice of minimal or no tooth preparation is a key factor in prognosis [10]. Although initial reports described a non-preparation technique, such a design may have negative effects on gingival tissue health resulting from over-contouring and a bulky margin, which restricts cleaning to create an unnatural gingival profile. In addition, gingival recession is also a common problem that has led experts to believe that this approach is compromised [5,11]. Moreover, the four basic conventional types of non-overlapped (window and feathered-edge) and overlapped (butt-joint and palatal chamfer) preparations optimize the bulky profile and overall contour and provide a definite finishing line, but the thickness of the preparation requires the unnecessary removal of sound tooth structure, exposing the dentin, and thereby reducing the bonding values and causing flexion in the tooth structure [12,13]. Minimally invasive veneer preparation designs, which have recently become popular, have also been described as an ultraconservative approach that involves less tooth reduction with thicknesses of 0.3–0.5 mm [11]. The suggested design aims to fulfill the three main criteria of strength, fit, and esthetics by overcoming the over-contouring and gingival problems of the non-preparation design as well as preserving as much of the tooth structure as possible, unlike the conventional design.

Although veneers are widely used as a conservative method, they have several drawbacks such as microleakage, marginal discrepancies, postoperative sensitivity, fractures, periodontal or pulpal disease, and debonding [14]. The most common failures associated with veneers are fracture and marginal defects, which are observed in 67% of cases after 15 years of clinical performance [15]. The high rates of failure are related to the amount and angle of stresses during mastication, cement polymerization shrinkage, and exposure to different thermal loads [16,17]. Dental preparation has also been considered as a factor influencing the failure of veneers [18].

Since certain variables in biomedical devices cannot be developed, tested, and monitored in living subjects using in vivo models because of ethical considerations and cannot be assessed in in vitro tests because of serious experimental difficulties [19], a multidisciplinary approach using finite element analysis (FEA) is being increasingly used since it offers an alternative approach to overcoming these problems. As a result, FEA has become an effective tool for visualizing and modeling the biomechanical characteristics of various dental and biological structures, thereby limiting the costs and risks involved in real-life experience [17,20]. Besides, this approach allows an estimation of the failure risk of unsuccessful dental treatment instead of experiencing this in clinical practice [21].

This study was initiated by realistic cone-beam computed tomography (CBCT) and accurate 3D solid geometry modeling of the anatomical dimensions and complex geometries of the maxillary tooth to provide more reliable and precise predictions, since most previous FEA studies have worked with a simplified 2D geometry or have used artificial teeth in a block model. The main aim of the present study was to investigate the influence of different preparation designs on the biomechanical behavior of anterior laminate veneers with different masticatory loads. Additionally, transient thermal analysis was employed for various thermal loading conditions to assess the effects of temperature variations on pulpal health and various tooth tissues and to assess the effect of the resultant thermal stresses on the long-term survival of veneer restorations using the 3D finite element method.

## 2. Materials and Methods

### 2.1. 3D-FEA Design of Model Geometry

A CBCT scan of a 28-year old clinically female with normal occlusion from an already available database, which was taken with NewTom Giano HR (Cefla, Imola, Italy) for other clinical purposes, was used for the reconstruction of a 3D finite model simulating the cross-section of a maxillary right central incisor. This scan was defined as the model for this analysis, as data from CBCT can be used to reconstruct 3D tooth geometry with high linear, volumetric, and geometric accuracy [22]. A total of 185 layered slice images with a voxel dimension of 68 mm, a field of view of 11 × 8 cm, and a slice thickness of 0.3 mm were saved in DICOM format and then imported into Mimics software (version 21.0, Materialise, Leuven, Belgium) for the construction of the surface and segmenting of the scanned objects into separate elements.

Global image thresholding was applied to reduce the size of the 3D geometry by selecting and sectioning the maxilla in the area related to the central incisor. To mimic the actual clinical conditions, all tooth parts, including the enamel, dentine, and pulp with the surrounding cementum and periodontal ligament, which plays a major role in distributing and transferring the heat and forces produced during thermal and masticatory function into the alveolar process through the bones [23], and the cortical and trabecular bone of the maxilla, which is important to prevent impractical stress distributions [24], were included in the 3D model, which were then saved in stereolithography (STL) format. The produced STL files were examined and assembled on Autodesk Meshmixer software version 3.0 (Autodesk, San Rafael, California), where mesh improvement and refinement were carried out.

The average geometrical dimensions of the solid tooth and its supporting structures were modified based on the anatomical measurements suggested by Wheeler's Dental Anatomy. The morphology of the enamel, dentin, and pulp was adjusted according to previous studies [25]. The thickness of cementum is considered to be 50 μm in the cervical area and increased gradually to 200 μm in the apical area, while the periodontal ligament covered 12.5 mm of the root surface with a thickness of 0.18 mm [26,27]. The supporting bone was divided into lamina dura, cancellous bone, and cortical bone, which were redesigned to standardize the geometry of each component [28], as shown in Figure 1.

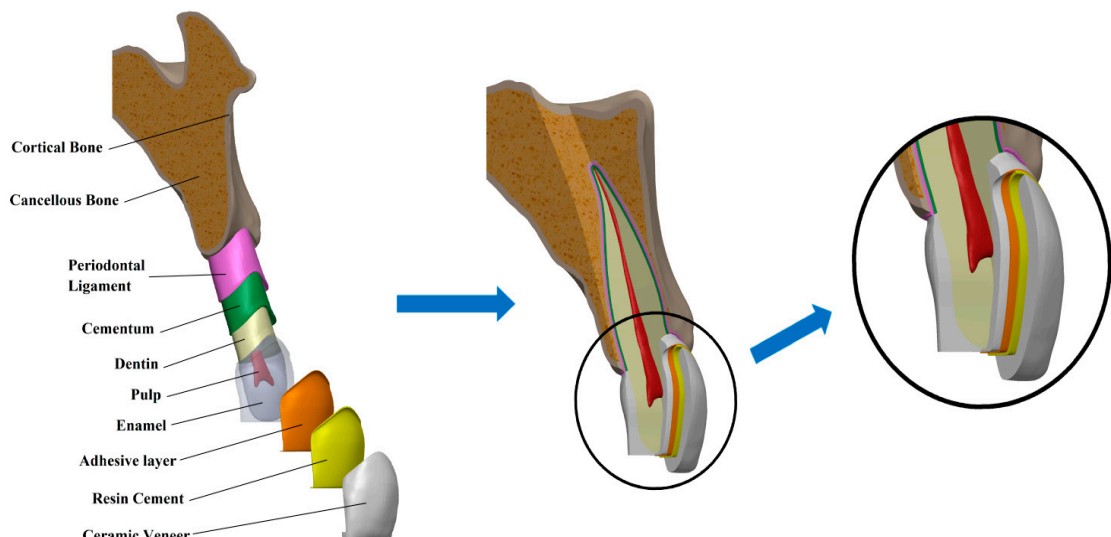

**Figure 1.** Exploded model and constructed geometry models of the maxillary central incisor with veneer restoration.

After the basic model of the central incisor was set up, a second modeling step was performed to obtain the veneer-restored incisors, based on the clinical presentation protocol specifying that the cervical margin is to be located 1.0 mm coronal to the cement-enamel junction. A distinct set of

procedures was followed to create the tooth pattern with three different veneer preparation designs as follows:

- Model A: Non-preparation design

Here, the laminate veneer was bonded directly on the unprepared tooth surface without any preparation of the underlying tooth structure or reduction of the incisal edge [1,5] (Figure 2A).

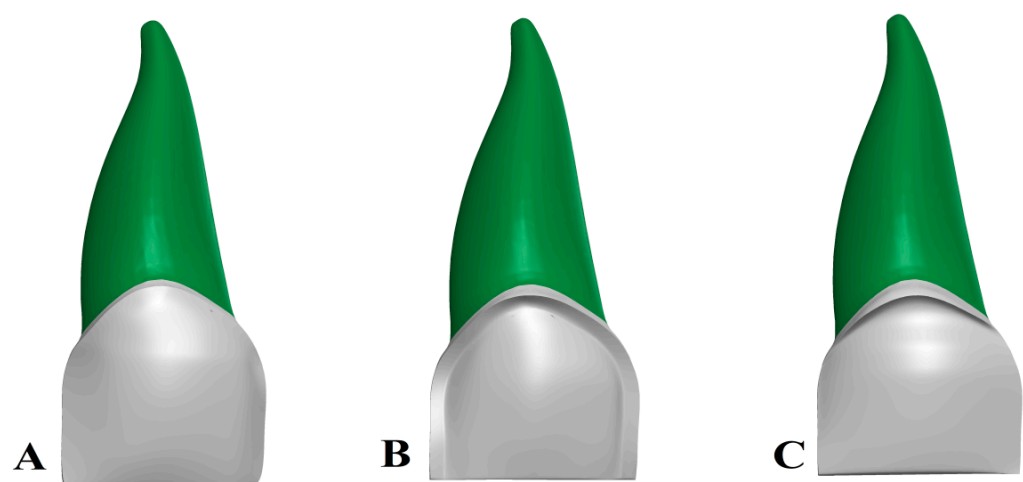

**Figure 2.** (**A**) Non-preparation design, (**B**) conventional preparation design, and (**C**) proposed preparation design.

- Model B: Conventional preparation design

The preparation was carried out according to the standard guidelines for tooth preparation required for veneers by making a chamfer finish line with a width of 0.5 mm at the cervical region on the labial surface and half of the proximal surfaces [7,16], and a 0.5-mm reduction on the labial surface in the middle and the incisal third continuing into the proximal area without breaking the interproximal contact area [6,29]. Moreover, preparations were made to the incisal edge (butt-joint design) in which the incisal edge was prepared bucco-palatally and reduced by 0.5 mm without involving the palatal area [29] (Figure 2B).

- Model C: Proposed preparation design

The proposed preparation design was modeled with a modification by combining the conventional preparation and non-preparation designs. This design involved creating a slight chamfer finish line with a depth of 0.5 mm at the level of the gingival margin on the labial surface and half of the proximal surfaces, and a 0.5-mm butt-joint incisal edge reduction as in the conventional preparation design, with no reduction in the middle and the incisal thirds of the labial surface of the tooth, similar to the non-preparation design (Figure 2C).

In this study, all models were restored with 0.5-mm zirconia-reinforced lithium silicate glass-ceramic (ZLS) (Celtra Duo - Sirona Dentsply, Milford, DE, USA). To bond the ceramic veneer on the enamel surface, resin luting cement of 100-μm thickness and an adhesive bonding agent of 50-μm thickness were applied to the inner surface of the veneer restorations shown in Figure 1, and each of these layers was created based on the physical dimensions given in the literature [30].

Next, the basic geometric models and the properties of all structures were imported into ANSYS software program. Each model was meshed to elements for FE simulations by a ten-node 3D tetrahedral structural solid with three degrees of freedom per node that showed a quadratic displacement behavior appropriate for irregular and complex geometries. To obtain the optimum number of elements, a standard convergence test analysis was carried out for all the models. The mesh was considered

acceptable since with reductions in the size of elements, the stress at the highest levels was similar to the results obtained with the previous mesh refinements, such that the final model precisely represented the original geometry [30,31] as shown in Figure 3 The total number of elements and nodes for the smallest mesh size used in our simulations were 481,917 and 111,655 for the non-prepare design, 507,560 and 125,383 for the conventional design, and 504,351 and 124,462 for the proposed design, respectively.

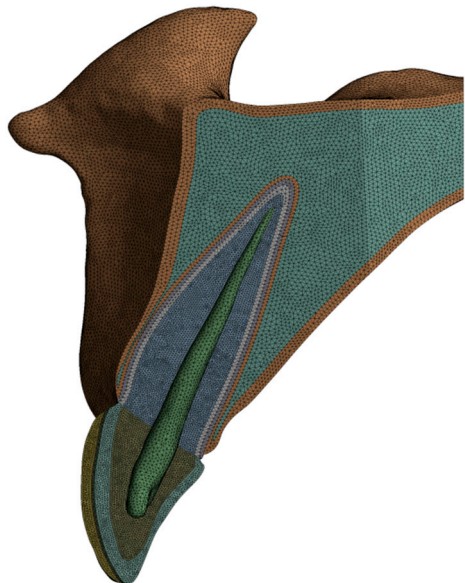

**Figure 3.** Sagittal section view of the finite element mesh of the maxillary central incisor with veneer restoration.

## 2.2. Biomechanical Stress Analysis

To demonstrate some of the in vivo masticatory forces that have a significant effect on the survival of veneers, two different loading conditions were investigated to detect the effects of the loading direction on each layer. A 100 N biting force was applied on the palatal surface of the crown at a point 1 mm from the incisal edge with the incisor longitudinal axis at an angulation of 60° and 125° to simulate intercuspal and protrusive movements, respectively [9,13]. To prevent relative motion at the interface in all models, the node at the base of the alveolar bone was set as the fixed support in all directions (x, y, and z) as a boundary condition. Then, static structural analysis was carried out to assess the von Mises equivalent stress distributions in the critical region of the veneer–cement layer–tooth interface [7].

## 2.3. Transient Thermal Finite Element Analysis

Transient thermal FEA was performed to determine the effects of extreme temperature changes on the pulpal and tissue health of the tooth and the effect of the resultant thermal stress distributions on the longevity of the veneer models during the simulated intake of hot and cold loads in the mouth, which present a time-dependent behavior.

An initial temperature of 36.5 °C was set as the base temperature for the whole model of the tooth to simulate the uniform environmental temperature in the mouth [17], after which analysis was performed by exposing the outer surface of the crown (veneer, enamel) of each model to a high temperature of 55 °C within a 2-s dwell time, which corresponds to the normal duration of exposure to hot thermal loading. Next, the temperature was relaxed and returned to the normal oral cavity temperature of 36.5 °C within 5 s. For the cold thermal loading, the same procedure was performed for a low temperature of 5 °C. These thermal loadings and time intervals were established based on previous studies in the literature [17,30].

The following thermal intake energy for the boundary conditions between the substance and the outer surface of the veneer was assumed:

- The convection film coefficient was taken as $5 \times 10^{-4}$ [J/(s mm °C)] for the 55 °C load, corresponding to the ingestion of hot liquids [32].
- The convection film coefficient was taken as $7.37 \times 10^{-4}$ [J/(s mm °C)] for the 5 °C load, corresponding to the ingestion of cold liquids [19].

All materials and anatomical structures were assumed to be homogeneous, linearly elastic, and isotropic. Since the analysis was performed for two assessments (mechanical and thermal), Young's modulus, Poisson's ratio, and the thermal expansion coefficient as well as density, thermal conductivity, and specific heat were defined for each part, according to the majority of studies that have employed FEA, as shown in Table 1.

**Table 1.** Mechanical and thermal properties of each material and tooth structure.

| Material | Young's Modulus (GPa) | Poisson's Ratio | Density (g/cm$^3$) | Specific Heat (J/(g °C)) | Thermal Expansion (1/°C) | Thermal Conductivity (J/(s mm °C)) | References |
|---|---|---|---|---|---|---|---|
| Enamel | 84.1 | 0.33 | 3 | 0.754 | $1.70 \times 10^{-5}$ | $0.92 \times 10^{-3}$ | [33,34] |
| Dentin | 18.6 | 0.31 | 2.20 | 1.172 | $1.06 \times 10^{-5}$ | $0.63 \times 10^{-3}$ | [33,34] |
| Cementum | 15.5 | 0.31 | 2.06 | 0.824 | $1.1 \times 10^{-5}$ | $0.62 \times 10^{-3}$ | [35] |
| Periodontal Ligament | 0.069 | 0.45 | 1.1 | 2.290 | $1.06 \times 10^{-5}$ | $0.59 \times 10^{-3}$ | [23,30] |
| Cancellous Bone | 1.37 | 0.30 | 0.62 | 1.16 | $1.0 \times 10^{-5}$ | $0.39 \times 10^{-3}$ | [4] |
| Cortical Bone | 13.7 | 0.30 | 2.06 | 1.26 | $1.0 \times 10^{-5}$ | $0.38 \times 10^{-3}$ | [4] |
| Pulp | 0.02 | 0.45 | 1 | 4.2 | $1.81 \times 10^{-5}$ | $0.63 \times 10^{-3}$ | [17,23] |
| Adhesive Layer | 4.5 | 0.3 | 2.02 | 0.824 | $1.06 \times 10^{-5}$ | $0.4 \times 10^{-3}$ | [33] |
| Luting cement | 8.3 | 0.24 | 1.1 | 0.824 | $3.0 \times 10^{-5}$ | $1.091 \times 10^{-3}$ | [7,30] |
| Celtra Duo (Dentsply) | 70 | 0.22 | 2.6 | 0.973 | $1.18 \times 10^{-5}$ | $1.463 \times 10^{-3}$ | Dentsply, manufacturer |

## 3. Results

### 3.1. Mechanical Loads

The maximum von Mises equivalent values recorded for all three preparation designs with two different loading directions in MPa are illustrated in Table 2 and Figures 4–6.

**Table 2.** Peak von Mises stress values (MPa) for all tested models with varying preparation designs and loading angles.

| Preparation Designs | Load Angle | Ceramic Veneer | Cement Layer | Tooth Structures | | | | |
|---|---|---|---|---|---|---|---|---|
| | | | | Enamel | Dentin | PDL | Cementum | Bone |
| Non- preparation (Mod A) | 60° | 75.96 | 42.73 | 26.87 | 20.87 | 15.34 | 12.31 | 54.40 |
| | 125° | 107.88 | 51.34 | 43.62 | 29.27 | 17.22 | 15.41 | 62.28 |
| Conventional (Mod B) | 60° | 122.99 | 31.99 | 33.16 | 21.90 | 17.29 | 13.15 | 55.99 |
| | 125° | 149.31 | 44.03 | 40.69 | 30.61 | 17.39 | 16.47 | 62.09 |
| Proposed(Mod C) | 60° | 64.87 | 32.01 | 23.45 | 19.54 | 15.10 | 12.359 | 55.57 |
| | 125° | 81.12 | 44.98 | 35.07 | 29.65 | 18.15 | 16.53 | 61.73 |

The stress distributions for the veneer restorations are shown in Figure 4. For intercuspal movement, the highest von Mises stress values recorded was 122.99 MPa for Mod B in the incisal margin, while Mod A showed a significantly lower von Mises stress value of 75.96 MPa concentrated at the cervical edge, and the lowest von Mises stress value recorded was 64.87 MPa for Mod C at the flat incisal area. A progressive increase was observed in the stresses when the angulation of the applied force was increased. The stress distribution in the ceramic veneer under protrusive movement achieved a maximum concentration of 149.31 MPa in the incisal area in Mod B, while in Mods A and C, the maximum values were 107.88 and 81.12 MPa in the cervical third and incisal bevel, respectively.

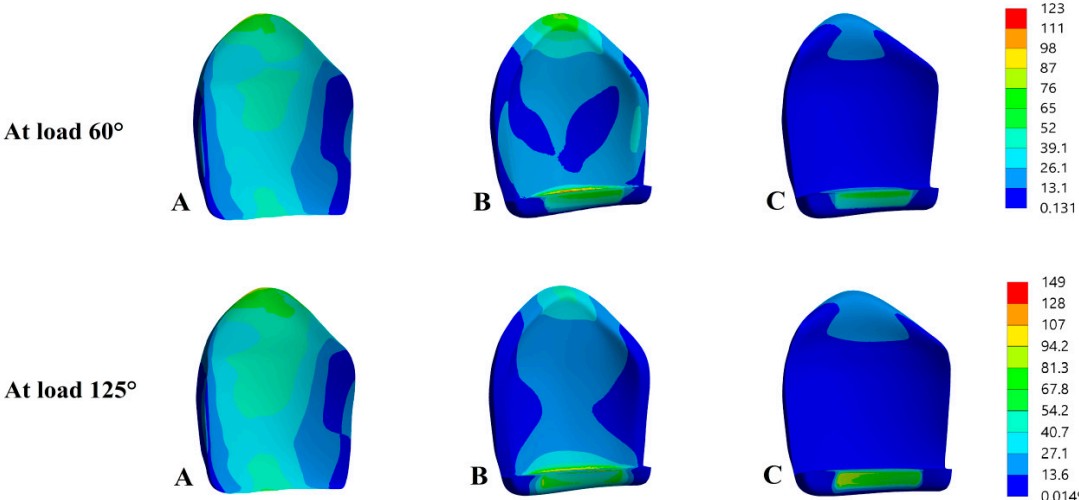

**Figure 4.** von Mises stress distributions (MPa) in the ceramic veneer. (**A**) Non-preparation design, (**B**) conventional preparation design, and (**C**) proposed preparation design.

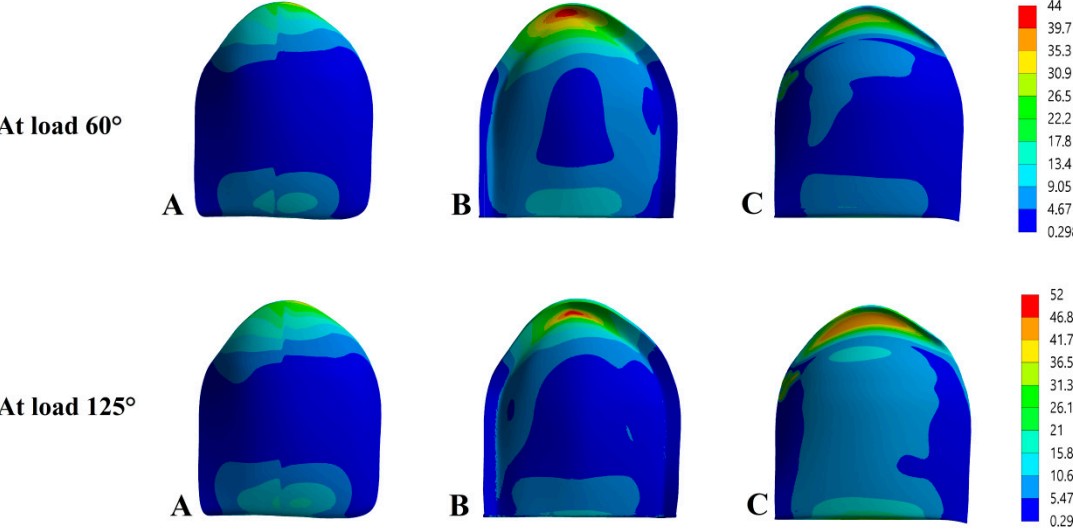

**Figure 5.** von Mises stress distributions (MPa) in the cement layer. (**A**) Non-preparation design, (**B**) conventional preparation design, and (**C**) proposed preparation design.

Stress concentrations in the cement layer were similar to those in veneer, and the values under the 60° and 125° forces are compared in Figure 5. For the 60° load, the maximum value observed in Mod A was 42.73 MPa, which occurred along the cervical line, while the stress values for both Mods B and C were very similar and gradually and significantly decreased to 31.99 and 32.01 MPa, respectively, in the center of the cervical region. For the 125° load, almost the same high value of 51.34 MPa was observed in Mod A, and both Mods B and C showed nearly identical values ranging between 44.03 and 44.98 MPa, respectively, in the cervical region.

The stress distribution in the tooth structures are shown in Figure 6. All models show symmetric behaviors for the distribution of stress on each part of tissue, although differences in the values were found in enamel. Under a protrusive load, the highest stress values of 40.69 and 43.62 MPa were obtained in Mods B and A in the cervical area buccally and palatally, with the lowest value being 35.07 MPa in Mod C recorded on the buccal aspect of the cervical edge. In contrast, the scale values for stress were noticeably reduced under intercuspal load to reach a maximum value of 33.16 MPa in Mod B followed by 26.87 MPa in Mod A, with the lowest value of 23.45 MPa in Mod C. For all models, the stresses were concentrated on the buccal and palatal aspects in the cervical region.

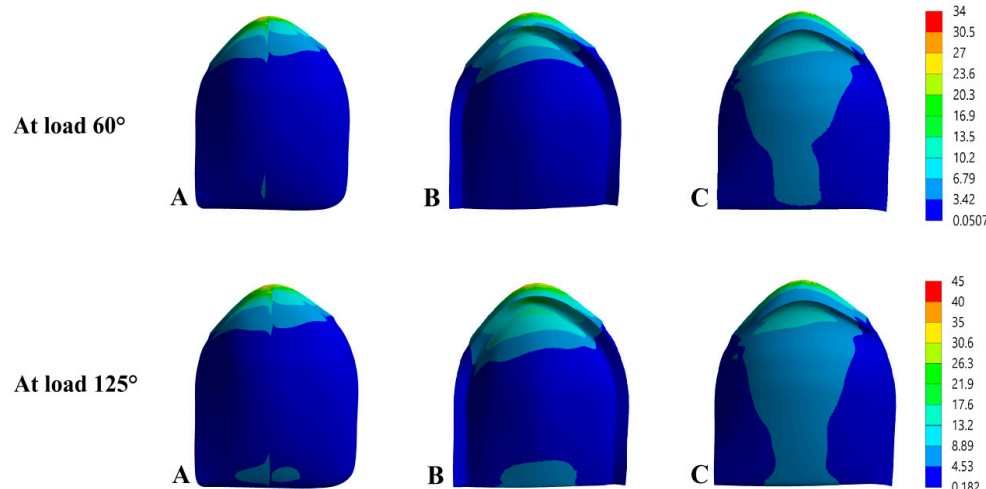

**Figure 6.** von Mises stress distributions (MPa) in tooth structure. (**A**) Non-preparation design, (**B**) conventional preparation design, and (**C**) proposed preparation design.

### 3.2. Transient Thermal Loads

#### 3.2.1. Temperature Distribution

In this section of the study for our simulations, four data points, A, B, C, and D on laminate veneer, enamel, dentin, and pulp were selected along the labio-palatinal (x-x) and the inciso-cervical (y-y) axes to represent the variations in temperature distributions in veneer restorations and tooth-supporting tissues for all design models (Figure 7). The results of the temperature distribution from the transient thermal analysis after exposure to 55 °C hot and 5 °C cold thermal loads are presented in Figures 8–10.

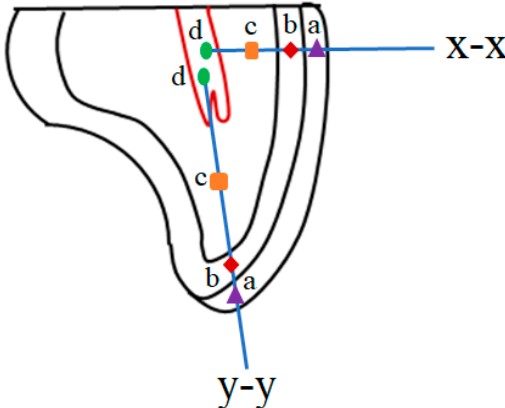

**Figure 7.** Calculated temperature change at the 4 points along both the x-x and y-y directions. (**a**) Veneer, (**b**) enamel, (**c**) dentin, and (**d**) pulp.

The temperature distributions in the non-preparation design are shown in Figure 8. As a result of exposure to the 55 °C hot load, the temperature value at the initial contact section, point A, increased to 45.8 °C in the labio-palatinal and 47.6 °C in the inciso-cervical directions. The temperature values gradually decreased to 44.1 °C at point B, 42.6 °C at point C, and 40.3 °C at point D through the labio-palatinal axis. In contrast, the maximum temperatures through the inciso-cervical axis at points B, C, and D were found to be 44.6°C, 40.6 °C, and 39.5 °C, respectively. For the 5 °C cold load, a temperature decrease was observed at point A to 21.8 °C in the x-x direction and 14.8 °C in the y-y direction. In contrast, the minimum temperature through the labio-palatinal direction at points B, C, and D reached 28.8 °C, 29.6 °C, and 30.9 °C, respectively, while thermal values at the same points through the inciso-cervical direction were 19.1 °C, 26.7 °C, and 30.1 °C, respectively.

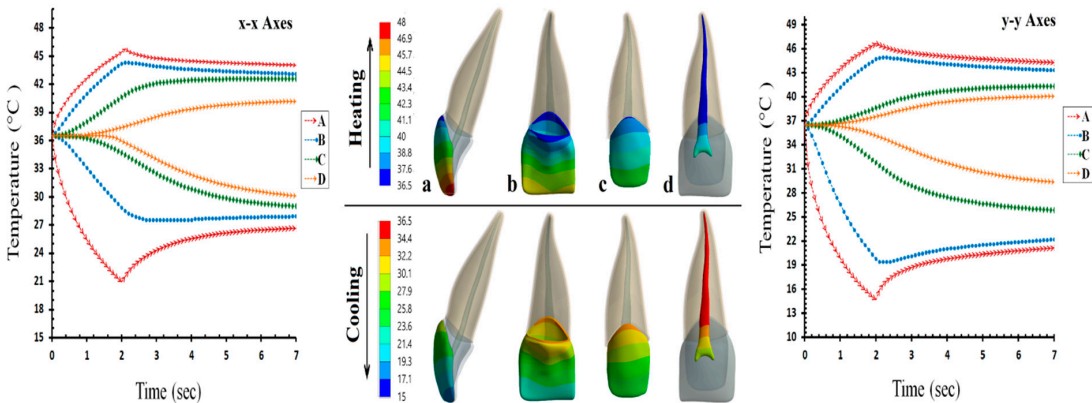

**Figure 8.** Temperature distributions in the non-preparation design along the x-x and y-y directions. (**a**) Veneer, (**b**) enamel, (**c**) dentin, and (**d**) pulp.

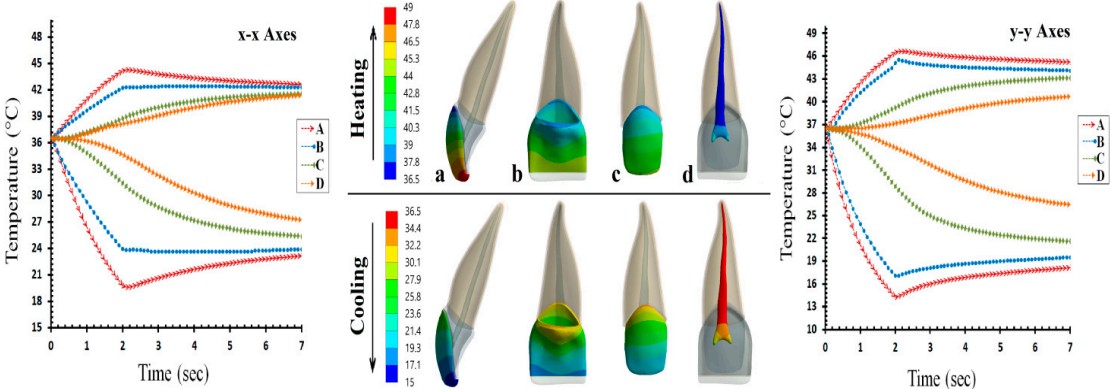

**Figure 9.** Temperature distributions in the conventional preparation design along the x-x and y-y directions. (**a**) Veneer, (**b**) enamel, (**c**) dentin, and (**d**) pulp.

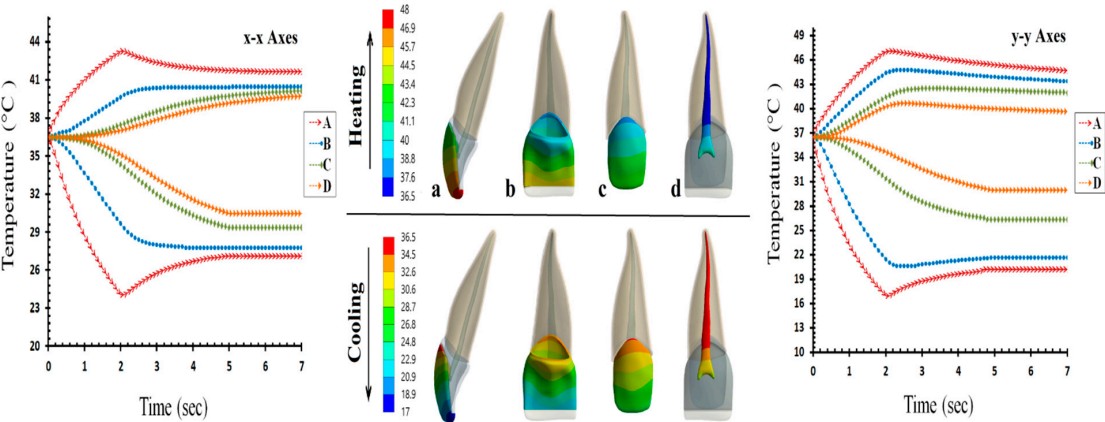

**Figure 10.** Temperature distributions in the proposed preparation design along the x-x and y-y directions. (**a**) Veneer, (**b**) enamel, (**c**) dentin, and (**d**) pulp.

The temperature distributions in the conventional preparation design are presented in Figure 9. Under the hot condition, the highest temperature along the x-x axis was 44.6 °C, which was observed at point A and that along the y-y axis was 47.4 °C. These temperature values decreased to 42.2 °C, 41.7 °C and 41.5 °C at points B, C, and D, respectively, through the x-x direction and 45.1 °C, 42.9 °C, and 41.8 °C, respectively, at the same locations through the y-y direction. In addition, under the cold condition, the lowest temperatures in both x-x and y-y axes were 19.7 °C and 14.2 °C, which were also

observed at point A. These temperatures were found to be 24.1 °C, 25.3 °C, and 27.2 °C at points B, C, and D, respectively, through the x-x direction and to 17.0 °C, 21.4 °C, and 26.9 °C, respectively, at the same points through the y-y direction.

Finally, the temperature changes in the proposed preparation design are shown in Figure 10. For hot loading, the maximum temperature was found to reach approximately 43.2 °C and 47.8 °C through the x-x and y-y directions on the outer ceramic point A. Along the inciso-cervical axis, these temperatures decreased to 40.2 °C for enamel at point B, 39.7 °C for dentin at point C, and 39.2 °C for pulp at point D, whereas temperatures of 44.8 °C for enamel, 42.8 °C for dentin, and 40.1 °C for pulp were obtained along the labio-palatal axis. For cold loading, the restoration temperatures decreased to 24.08 °C and 17.96 °C at point A along the x-x and y-y directions, respectively. The thermal results at points B, C, and D were 27.7 °C, 28.7 °C, and 31.1 °C in the bucco-palatinal direction and 21.4 °C, 27.4 °C, and 31.2 °C in the inciso-cervical direction.

### 3.2.2. Stress Distribution

In this part, von Mises thermal stress distributions caused by temperature changes on the ceramic veneer, resin cement, and an adhesive layer in each of the three models were studied for a duration of 2 s, as shown in Figure 11.

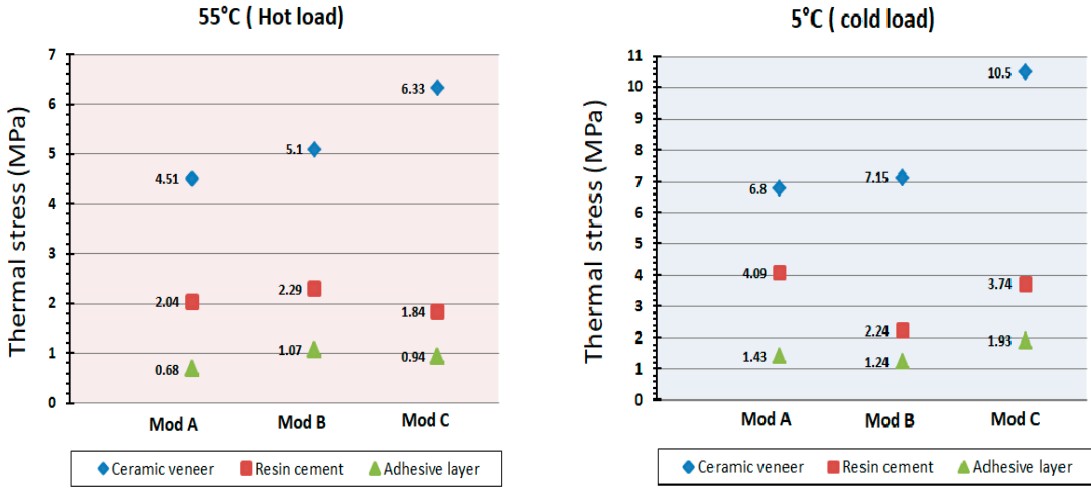

**Figure 11.** Maximum von Mises thermal stresses (MPa) in three layers of the components after application of cold (5 °C) and hot (55 °C) thermal loads.

The thermal stresses at each of the three layers after consumption of 55 °C hot liquid are shown in Figure 12. The maximum stress concentrations appeared in the ceramic veneer in the center of the incisal edge in Mod A, at the distal incisal border in Mod B, and the mesial incisal edge in Mod C in the range of 4.51 MPa, 5.1 MPa, and 6.33 MPa, respectively. The stresses then dropped significantly toward the cement layer, reaching a maximum value of 2.04 MPa in Mod A, 2.29 MPa in Mod B, and 1.84 MPa in Mod C and were concentrated in the incisal region mesially for all models. The adhesive layer showed smaller stress values that varied between 0.68 MPa, 1.07 MPa, and 0.94 MPa in the cervical third distally, at the incisal border mesially, and at the mesial cervical line in Mods A, B, and C, respectively.

The thermal distributions in all three models after the intake of 5 °C cold liquid are shown in Figure 13. For Mod A, the highest stress of 6.8 MPa occurred on the veneer at the center of the incisal bevel, followed by stress concentrations of 4.09 MPa on the luting cement at the mesial incisal edge and 1.43 MPa on the adhesive layer at both mesial and distal cervical edges. In contrast, the maximum stresses on the restoration, resin cement, and the adhesive layer for Mod B were 7.15 MPa, 2.24 MPa, and 1.24 MPa, respectively, in the cervical edge mesially and distally for all the layers. On the other

hand, the maximum stresses for Mod C were 10.5 MPa, 3.74 MPa, and 1.93 MPa in the incisal edge distally and mesially for the same components, respectively.

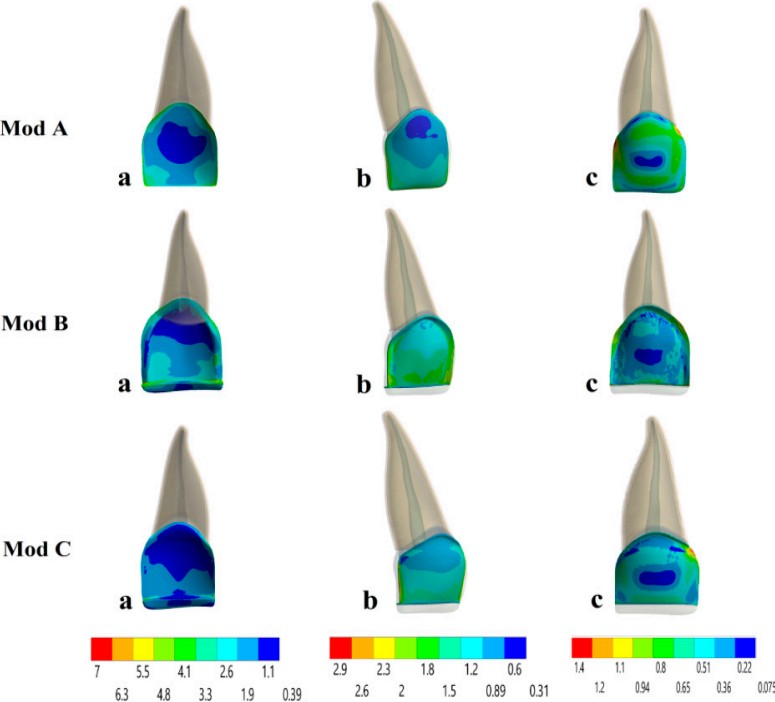

**Figure 12.** von Mises thermal stress distributions (MPa) in the non-preparation (Mod A), conventional preparation (Mod B) and proposed preparation (Mod C) models due to the hot thermal loading 55 °C on the (**a**) ceramic veneer, (**b**) resin luting cement, and (**c**) adhesive layer.

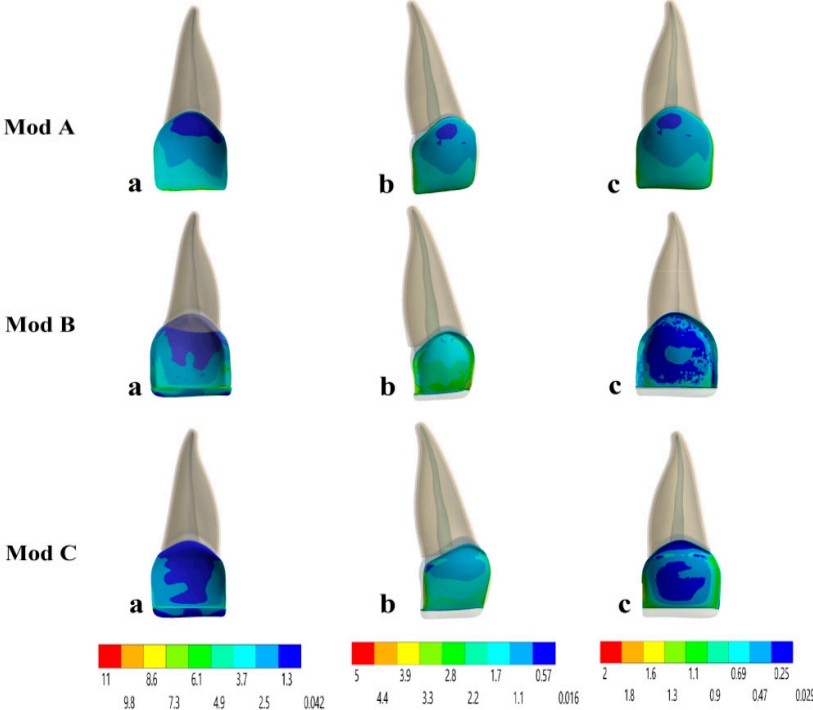

**Figure 13.** von Mises thermal stress distributions (MPa) in the non-preparation (Mod A), conventional preparation (Mod B) and proposed preparation (Mod C) models due to the cold thermal loading 5 °C on the (**a**) ceramic veneer, (**b**) resin luting cement, and (**c**) adhesive layer.

## 4. Discussion

Esthetic treatment of anterior teeth continues to be a challenge in clinical practice. At present, porcelain laminate veneers are steadily becoming more popular as a successful technique for restoring discolored, worn, malformed, or fractured teeth [36]. However, the geometry of the preparation for laminate veneers is a controversial topic. Although several studies have examined the influence of prepared and unprepared veneer designs on the longevity and survival of veneer restorations, data concerning the design and thickness of the laminate veneer preparation are limited [7]. The present study attempted to study the effect of the proposed preparation design as a conservative treatment option that aims to preserve both gingival health and sound tooth structure and compared it with previously used techniques. It used the 3D finite element method to investigate the amount and distribution of stresses under different masticatory and thermal loading conditions in order to understand the sites where stresses and damage to the veneer restorations and tooth structures occur in clinical observations.

In the mechanical loading assessments, in all of the studied preparation designs, the maximum level of the stress distribution was concentrated on the veneer surface and it progressively decreased toward the cement layer up to the dental tooth structure. This is because zirconia-reinforced lithium silicate is known to have a higher modulus of elasticity than both the resin cement and the tooth structure, indicating that ceramics act as a barrier during functional movements by absorbing most stresses and protecting underlying dental tissues; this finding is in agreement with previous studies [7,10].

In the evaluation of stress values in relation to the three layers that were analyzed (veneer, resin cement, and tooth structures), the conventional design showed the highest von Mises stresses at the level of the incisal region, followed by the non-preparation design, which showed moderate stresses distributed in the cervical areas, while the proposed technique showed the lowest stresses at the interface between the three layers condensed in the incisal and cervical areas. The resulting stresses were higher at 125° than at 60°, which agreed with the findings of previous studies [7,16], indicating that protrusive movement causes higher stresses than intercuspal movement. On the basis of the above outcomes, all models may result in failure in the form of fracture, chipping, or debonding at the incisal margin and cervical area, which agrees with a study performed by Li et al., who concluded that most veneer failures occurred in the incisal and cervical regions [13]. Thus, the probability of failures will increase with an increase in the angulation from 60° to 125° due to increases in the tensile stresses, in agreement with other studies [10,37] in which the fracture potential of ceramic material was shown to fail in areas of tension because of the low tensile strength of the material.

Based on the mechanical results, the proposed preparation design can be considered as an acceptable alternative approach for restoring anterior defects with laminate veneer since it shows the lowest von Mises values in veneer and tooth structures, a more uniform stress distribution in cement and tooth tissues, and a highly promising clinical success rate. The findings showing that the preparation design has a significant effect on the treatment success of veneer restorations are in agreement with the results of a study by Cotert et al. [18].

Thermal heat transfer in human teeth is a common process in both daily life and clinical dentistry. Accordingly, transient thermal FE simulation was performed to assess the effect of preparation geometries, temperature variations, and the resultant stresses on the structural changes in dental hard tissues and pathological changes in pulpal soft tissue. Therefore, maintenance of pulp health and dentin safety is a fundamental challenge; in addition, it is important to determine the critical temperature at which damage may occur. Zach and Cohen stated that if the temperature changes within the pulpal chamber exceed the critical limit of 5.5 °C, these changes will result in pulp necrosis. This critical threshold has been considered in many reports and studies [38].

Overall, the results of the temperature distribution at the outer surface were different from those obtained inside tooth structures in both vertical and horizontal directions. In accordance with our simulation results, thermal changes promoted by cold loading simulations were identified to be more effective than those induced by hot loading simulations. This is mainly due to the transfer of heat

by convection to the outer surface, in which cold thermal loading shows a larger heat convection coefficient value and temperature difference than hot loading, which is in good agreement with the literature [19,39].

The highest pulp temperature was seen in the conventional design, which was considered to be near the critical temperature for pulp injury, while in dentin, the temperature change for cold loading was below the physiological limit. This thermal change may induce tooth thermal sensitivity as a result of dentinal fluid flow (DFF) in the dentine tubules that can transmit a considerable amount of heat to the pulp chamber causing sensitivity; in addition, the pulpal blood flow (PBF) rate decreases during cooling [40]. While the thermal temperature changes in both the non-preparation and proposed preparation designs were within the physiologic limits and were not strong enough to harm the pulpal nerves or promote dentin sensitivity compared to the conventional design, these differences were due to less removal of dentin and enamel during preparation, in addition to the poor thermal conduction properties of the dentin microstructure, which is in good agreement with the outcomes observed previously [40,41].

Thermal variations in the oral environment are ultimately responsible for stresses that have an important effect on the lifespan of restorations by inducing microleakages, chipping, cracks, and fractures. Based on our results, all the models exhibited the maximum von Mises stress values in each studied layer under the cold thermal loading, and these values were higher than those under the hot thermal loading. In the assessments of the thermal stresses, the largest stress was concentrated on the veneer surface, which is the area in contact with the thermal load. Among the tested models, the proposed preparation design showed relatively lower von Mises thermal stress values than the conventional and non-preparation designs, which were more durable under these thermal conditions.

Additionally, stress distribution in the incisal areas is higher than that in the cervical areas, which may promote chipping and fracture in the veneer; furthermore, stresses on palatal surfaces were higher than those on the buccal surfaces. This action is based on the high elastic modulus and the coefficient of thermal expansion properties of ceramic veneer, which therefore exhibited higher thermal stresses. The accumulation of thermal stresses in the bonding layer may lead to inner flaws in the cement layer, which cause crack propagation and microleakage. Furthermore, polymerization shrinkage, function, and thermocycling contribute to premature debonding, and the intrinsic tensile stress contributes to the damage to the ceramic veneer. These findings were in agreement with those of previous studies [19,30].

As FEA is a numerical tool, some aspects of the oral environment, such as the non-linear elastic characteristics of the PDL tissues and the anisotropic feature of dentin were not considered since they require substantial experimental data. In addition, it is essential to perform a long-term clinical and experimental study to assess the ultimate clinical efficacy of these designs and to obtain proof to make clinical decisions since no direct clinical data on this issue is yet available.

## 5. Conclusions

Within the limitations of this study, the following conclusions can be drawn:

1.  The preparation design has a significant effect of the mechanical behavior of laminate veneer restorations, which influences the integrity and survival rate of teeth restored with laminate veneer restorations, and the most effective preparation design recommended in this study for ceramic veneers is the proposed design.
2.  Different angulations induce different patterns of stress concentration. The fracture potential of veneer restorations was higher and more non-uniform at an angle of 125° than at 60°.
3.  In all types of preparation designs, the temperature changes were within the physiologic limits without enhancing any risk to the supporting hard tissues and dental pulp.
4.  Cold thermal loading was much effective than hot thermal loading. Thus, cold simulations yield thermal stresses of higher magnitude, which increases the risk of failure and short-term success of laminate veneers.

The results and conclusions provided in this study suggest that the proposed preparation design provided a more appropriate geometry for stress distribution and preservation of the health of dental tissues in comparison with the other techniques, and indicate the clinical use of the laminate veneers with the proposed preparation as a restorative approach for the correction of deformed and unaesthetic teeth, which was found to be promising and safe, preventing short-term failure under functional and thermal conditions.

**Author Contributions:** Conceptualization, S.N.D. and A.R.A.-Z.; Data curation, L.A.S.; Formal analysis, L.A.S.; Investigation, S.N.D.; Methodology, S.N.D.; Software, L.A.S.; Supervision, A.R.A.-Z.; Validation, S.N.D.; Writing—original draft, S.N.D.; Writing—review & editing, A.R.A.-Z. All authors critically revised the manuscript and agreed to the published version of the manuscript.

**Funding:** This research received no external funding.

**Acknowledgments:** This study is one part of the Ph.D. thesis at the Department of conservative dentistry, College of Dentistry, University of Sulaimani, and supported in part by the Ministry of Higher education, the author also acknowledges the Dental Center for providing CBCT scan with mimic image processing software.

**Conflicts of Interest:** The authors declare that they have no conflicts of interest.

**Ethical Approval:** This chapter does not contain any studies with human participants or animals performed by any of the authors, and the image of CBCT used was taken from a publicly available database.

**Data Availability:** The data used to support the findings of this study are available from the corresponding author upon request.

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
