# Peer review of "Mechanical and Thermal Stress Behavior of a Conservative Proposed Veneer Preparation Design for Restoring Misaligned Anterior Teeth: A 3D Finite Element Analysis"

_applsci, doi:10.3390/app10175814_

Round 1

Reviewer 1 Report

The manuscript submitted presents the results of an modelling study, to assess the impact of preparation design on the biomechanical behavior on laminate veneers, and the impact of cold and warm loading.

The problematic is clearly presented in the Introduction section.

The methodology is well explained, in term of tooth preparation design and Finite Element Analysis.

The discussion is smartly based on the results, with permanent comparison with previous studies.

The temperature loading part sounds a bit less instructive than the mechanical behavior part, but, as the whole FEA model was ready, it makes sense to include it in the whole study.

The overall work gives precious evidence-based elements to justify minimal tooth preparation for laminar veneers restoration.

Here are some side remarks/questions :

1/ Fig. 8 – 9 – 10

How do the authors explain the difference in temperature reaction of veneers (especially cold loading) between non-preparation design, conventional preparation and the proposed preparation design ? (21°C / 19°C / 24°C).

This point is difficult for us to understand, as the same ceramic material is used.

2/ Fig. 8 – 9 – 10

How do the authors explain that after the 2 sec temperature loading, the assessed temperature starts to go back to initial value (36.5°C) but rapidly reaches a plateau (stable value after 2-3 sec), and seems not to move anymore.

3/ Conclusion L. 408 : « The geometry of the preparation design has a significant effect on the integrity and survival rate of teeth restored with laminate veneer restorations ».

As this is a modelisation, it might be more correct to say that the preparation design has a significant effect of mechanical behavior of laminate veneer restorations, which will influence the integrity and survival rate…

4/ Further applications : Have the authors consider to use their modelisation to follow mechanical behavior of premolars or molars ?

Reviewer 2 Report

The paper entitled “Mechanical and Thermal Stress Behavior of a Conservative Suggested Veneer Preparation Design for Restoring Misaligned Anterior Teeth: A 3-D Finite Element Analysis” by Shilan Nawzad Dawood  et. al. used finite element analysis (FEA) to investigate stress and thermal load on three different porcelain veneer designs. CBCT images were used to reconstruct a maxillary incisor. Mechanical and thermal characteristics of tooth , supporting structures, veneers, cement and bonding were referenced from literature and ANSYS software was used for comparative studies of stress and transient thermal distribution.  Authors concluded teeth prepared with a proposed design showed lower stress distributions.

This is a very well written paper and results seem logically listed within the limitation of the experimental protocols.  I would like authors to clarify the following concerns:

  • CBCT was used to obtain slice images of maxillary incisors. What type of incisor was used? Central or lateral
  • It seems authors performed considerable post image processing in Meshmixer software to create an ideal tooth with its surrounding structures using Wheeler’s textbook. I was wondering why a CBCT scan was used at all.  An ideal tooth with its surrounding structures could be easily created in Meshmixer or similar 3D software, what was the purpose of using CBCT images?
  • Simulation model was designed to apply a 100 N biting force to the palatal surface of the crown near the incisal edge. Where was the force exactly applied. In at least two of the veneer preparation design , it appears that porcelain covers 0.5 mm of the incisal edge.  Was the force applied to the porcelain at the incisal edge directly? How far was the force application point from the porcelain/tooth interface?
  • Assumptions were made that all materials and anatomical structures were homogeneous, linearly elastic, and isotropic. Although, often such inaccurate assumptions are made to simplify the FEA analysis and interpretations. I was wondering if authors could extrapolate on a more realistic non-homogenous ,non linearly elastic and anisotropic model.
